# Center of Mass Estimation Using a Force Platform and Inertial Sensors for Balance Evaluation in Quiet Standing

**DOI:** 10.3390/s23104933

**Published:** 2023-05-20

**Authors:** Motomichi Sonobe, Yoshio Inoue

**Affiliations:** Department of Intelligent Mechanical Systems Engineering, Kochi University of Technology, Kochi 782-8502, Japan

**Keywords:** quiet standing, balance evaluation, center of mass, double-inverted pendulum, force platform, inertial sensor

## Abstract

Accurate estimation of the center of mass is necessary for evaluating balance control during quiet standing. However, no practical center of mass estimation method exists because of problems with estimation accuracy and theoretical validity in previous studies that used force platforms or inertial sensors. This study aimed to develop a method for estimating the center of mass displacement and velocity based on equations of motion describing the standing human body. This method uses a force platform under the feet and an inertial sensor on the head and is applicable when the support surface moves horizontally. We compared the center of mass estimation accuracy of the proposed method with those of other methods in previous studies using estimates from the optical motion capture system as the true value. The results indicate that the present method has high accuracy in quiet standing, ankle motion, hip motion, and support surface swaying in anteroposterior and mediolateral directions. The present method could help researchers and clinicians to develop more accurate and effective balance evaluation methods.

## 1. Introduction

Balance analysis during quiet standing has been used to evaluate stroke [1], Parkinson’s disease [2], dizziness [3], child development [4], and fall prediction in the elderly [5,6]. The center of pressure (COP), which is convenient to measure, is typically used for balance evaluation [7,8,9]. However, when considering the central nervous system as the control system, it is necessary to evaluate the relationship between the displacement and velocity of the center of mass (COM) as the input and ankle joint moment as the output [10,11]. Rather than ankle joint torque, it is possible to evaluate COM acceleration or COP proportional to ankle joint torque [12,13]. COM acceleration can be directly estimated from the horizontal force on the force platform [14]. Thus, while the COP and COM acceleration can be measured using a force platform, it is necessary to estimate the COM displacement and velocity using other balance evaluation methods.

COM estimation for balance evaluation requires measurement in a short time without extra effort and with high accuracy on the order of a few millimeters. Furthermore, a COM estimation method applicable to a moving support surface is desirable because external stimuli may be added to the support surface in some balance evaluation methods [15,16,17,18]. The gold standard for COM measurement is an optical motion capture system with reflected markers [11]. Despite its high accuracy, this method has not been used clinically because of labor and measurement uncertainties. For other COM estimation methods, we can use markerless motion capture systems (multi-camera systems or depth cameras such as Azure Kinect) or motion capture systems using wearable inertial measurement units (IMUs). However, to the best of our knowledge, no studies have applied these devices to the COM analysis of quiet standing owing to accuracy issues.

To satisfy practicality and accuracy requirements, COM estimation techniques based on force platform measurements have been studied [19]. These challenges can be roughly classified into the following three approaches: the first is to integrate the horizontal acceleration of the COM [20], the second is to filter the COP to estimate the COM [21], and the third is to estimate the COM directly from the equations of motion of an inverted pendulum model. The first and second approaches are based on a single inverted pendulum model, whereas the third method can be extended to more flexible models, such as a double inverted pendulum model. However, these methods have rarely been applied in balance evaluation for the following reasons. The first method of integrating horizontal forces is considered to have insufficient estimation accuracy because it primarily uses horizontal forces, which are difficult to measure. The second method, COP filtering, has the problem that the COP and COM are not independent, which makes it difficult to use for balance evaluation. The third method, which is based on equations of motion, has not been extended to more complex models than a single inverted pendulum, and the accuracy verification is insufficient.

Mechanical models are widely used in the analysis of any standing posture and not only for COM estimation [22,23]. Inverted pendulum models are typically assumed, where the foot is fixed to a support surface, and the segments above the ankle have degrees of freedom. Typical models include a single pendulum model consisting of the foot and body; a double pendulum model consisting of the foot, lower body, and upper body; and a triple pendulum model consisting of the foot, lower leg, thigh, and upper body. Although a simpler model allows COM estimation with less information, it tends to have larger estimation errors. The validity of the single inverted pendulum model to represent COM during standing is divided into positive [24] and negative [25] arguments and seems to depend on the required accuracy and the subject’s motion. For example, it has been shown that large errors occur in COM estimation when subjects perform hip strategy motion [26]. However, because increasing the degrees of freedom of the model requires more measurement information for COM estimation, it is desirable to accurately estimate the COM for all motions with a model with fewer degrees of freedom.

In this study, we aimed to estimate the COM displacement and velocity in the sagittal and frontal planes based on the equations of motion. We used a force platform as the primary measurement device and IMUs to extend the model. To investigate the estimation accuracy of the proposed method, we derived the error of the estimation method using the COM estimated from optical motion capture as the true value. For comparison, we also estimated the COM using the integration method of COM acceleration and the COP filtering method reported in previous studies. In this study, we validated the proposed method by applying the COM estimation method not only to quiet standing but also to strong hip strategy movements and standing on a moving support surface.

## 2. Estimation Methods

### 2.1. Modeling

In this study, the COM movement of a standing subject was measured by mounting a force platform on the support surface and attaching an IMU to the head. The horizontal sway of the support surface was measured using an acceleration sensor. The human body is comprised of multiple rigid bodies. We ignored the vertical motion of the human body and assumed that the variation in posture angle was small. The stationary coordinate system O-XYZ and moving coordinate system P-XYZ, whose origin is fixed to the center of the force platform, were defined to derive the equations of motion. The horizontal forward direction is the *x*-axis, the horizontal left-hand direction is the *y*-axis, and the vertical upward direction is the *z*-axis.

The simplest model, composed of the body and foot in the sagittal plane, is shown in Figure 1a. This model assumes that the feet are fixed to the support surface, which can move horizontally, and that the body rotates around the ankle. When the posture angle of the body is small, the equations of motion for the model are as follows:(1)$$m_{b}{\ddot{x}}_{b}(t)=-M{\ddot{X}}_{s}(t)-R_{x}(t),$$
(2)$$\left\{\frac{J_{xb}}{l_{b}}+m_{b}(L_{f}+l_{b})\right\}{\ddot{x}}_{b}(t)-m_{b}gx_{b}(t)=-M(L_{f}+l_{b}){\ddot{X}}_{s}(t)-N_{y}(t),$$
where *x_b_* is the COM displacement in the moving coordinate system; *Ẍ_s_* is the acceleration of the support surface in the stationary coordinate system; *N_y_* and *R_x_* are the moment around the *y*-axis and the horizontal force in the *x*-axis measured by the force platform, respectively; *M* is the weight of a subject; *m_b_* is the body weight; *J_xb_* is the moment of inertia around the body COM; *l_b_* is the length between the ankle joint and the body COM; *L_f_* is the height of the ankle; and *g* is the gravitational acceleration.

As a more complex model describing the hip joint strategy motion, a double inverted pendulum model consisting of the foot, lower body, and upper body is shown in Figure 1b. This model allows for rotational motion of the lower and upper bodies around the ankle and hip joints. Assuming that these posture angles are small, the equations of motion are given by
(3)$$m_{1}{\ddot{x}}_{1}(t)+m_{2}{\ddot{x}}_{2}(t)=-M{\ddot{X}}_{s}(t)-R_{x}(t),$$
(4)$$J_{x1}{\ddot{x}}_{1}(t)+J_{x2}{\ddot{x}}_{2}(t)-m_{b}gx_{b}\;(t)=-M(L_{f}+l_{b}){\ddot{X}}_{s}(t)-N_{y}(t).$$
where
(5)$$\begin{array}{l} J_{x1}=\frac{J_{1}}{l_{1}}+m_{1}(L_{f}+l_{1})-\frac{J_{2}L_{1}}{l_{1}l_{2}},\\ J_{x2}=\frac{J_{2}}{l_{2}}+m_{2}(L_{f}+L_{1}+l_{2}). \end{array}$$

Here, *x*_1_ and *x*_2_ are the COM displacements of the lower and upper bodies in the moving coordinate system; *J*_1_ and *J*_2_ are the moments of inertia around the COMs of the lower and upper bodies, respectively; *L*_1_ is the length of the lower body; *l*_1_ is the length from the ankle joint to the lower body COM; *l*_2_ is the length from the hip joint to the upper body COM; and *x_b_* is the combined COM of the lower and upper bodies derived from *x_b_* = (*m*_1_*x*_1_ + *m*_2_*x*_2_)/*m_b_*.

A frontal plane model composed of both feet, both legs, the pelvis, and the upper body is shown in Figure 1c. The feet were fixed to the support surface, and the feet, legs, and pelvis were connected in a loop. To simplify the model, we assumed that the legs were parallel and that the boundary between the pelvis and the upper body was the waist joint around the fifth lumbar vertebra. This assumption prevents pelvic rotation. Assuming that the tilt angles of the legs and upper body are small, the equations of motion are given by
(6)$$(2m_{l}+m_{p}){\ddot{y}}_{l}(t)+m_{u}{\ddot{y}}_{u}=-M{\ddot{Y}}_{s}(t)-R_{y}(t),$$
(7)$$J_{y1}{\ddot{y}}_{l}(t)+J_{y2}{\ddot{y}}_{u}(t)-m_{b}gy_{b}(t)=-M(L_{f}+l_{b}){\ddot{Y}}_{s}(t)+N_{x}(t),$$
where
(8)$$\begin{array}{l} J_{y1}=\frac{2J_{l}}{l_{l}}-\frac{J_{u}L_{l}}{l_{u}l_{l}}+2m_{l}(L_{f}+l_{l})+m_{p}\frac{L_{l}}{l_{l}}(L_{f}+L_{l}+l_{p}),\\ J_{y2}=\frac{J_{u}}{l_{u}}+m_{u}(L_{f}+L_{l}+L_{p}+l_{u}). \end{array}$$

Here, *y_b_* is the combined COM displacement except for feet; *y_l_* is the COM of both legs; *y_u_* is the upper body COM in the stationary coordinate system; ${\ddot{Y}}_{s}$ is the acceleration of the support surface in the moving coordinate system; *N_x_* and *R_y_* are the moments around the *x*-axis and the horizontal force to the *y*-axis measured by the force platform, respectively; *m_l_* is the mass of one leg; *m_p_* is mass of the pelvis; *J_l_* is the moment of inertia of one leg around the COM; *J_u_* is the moment of inertia of the upper body around the COM; *l_l_* is the length between the ankle joint and the leg COM; *L_l_* is the length between the ankle joint and the hip joint; *l_p_* is the length between the hip joint and pelvis COM; *L_p_* is the length of the pelvis; and *l_u_* is the length between the waist joint and the upper body COM.

When the leg and upper-body angles are constrained to be equal, the equations of motion can be simplified as follows:(9)$$m_{b}{\ddot{y}}_{b}(t)=-M{\ddot{Y}}_{s}(t)-R_{y}(t),$$
(10)$$J_{yb}{\ddot{y}}_{b}(t)-m_{b}gy_{b}(t)=-M(L_{f}+l_{b}){\ddot{Y}}_{s}(t)+N_{x}(t),$$
where
(11)$$J_{yb}=\frac{2J_{l}+J_{u}+2m_{l}l_{l}(L_{f}+l_{l})+m_{p}L_{1}(L_{f}+L_{l}+l_{p})+m_{u}(L_{l}+l_{u})(L_{f}+L_{l}+L_{p}+l_{u})}{l_{b}}.$$

### 2.2. COM Estimation Method

The measurement devices used in this study were a force platform, an IMU attached to the back of the head, and an acceleration sensor on the support surface. From these devices, we can obtain the horizontal forces *R_x_* and *R_y_*, horizontal head acceleration *Ẍ_h_*, *Ÿ_h_*, and support surface acceleration *Ẍ_s_*, *Ÿ_s_*. In the following section, we demonstrate COM estimation methods for two cases: with and without a head IMU.

Without a head IMU, COM estimation methods are based on the simple inverted pendulum systems described in Equations (1), (2), (9), and (10). The COM displacement and acceleration are obtained by solving (1) and (2) for *ẍ_b_* and *x_b_* in the sagittal plane or by solving (9) and (10) for ${\ddot{y}}_{b}$ and *y_b_* in the frontal plane, respectively.

With a head IMU, the relationship between the head acceleration and COM acceleration of the lower and upper bodies can be described as follows:(12)$${\ddot{x}}_{h}(t)={\ddot{X}}_{h}(t)-{\ddot{X}}_{s}(t)=\frac{L_{1}}{l_{1}l_{2}}(l_{2}-L_{2}){\ddot{x}}_{1}(t)+\frac{L_{2}}{l_{2}}{\ddot{x}}_{2}(t),$$
(13)$${\ddot{y}}_{h}(t)={\ddot{Y}}_{h}(t)-{\ddot{Y}}_{s}(t)=\frac{L_{l}}{l_{l}l_{u}}(l_{u}-L_{u}){\ddot{y}}_{l}(t)+\frac{L_{u}}{l_{u}}{\ddot{y}}_{u}(t).$$
where *L*_2_ is the length between the hip joint and head IMU, and *L_u_* is the length between the waist joint and head IMU. In the sagittal plane, COM displacement of the body is estimated by solving the following equations derived from (3), (4), and (12) for *ẍ*_1_, *ẍ*_2_, and *x_b_*,
(14)$$\left[\begin{array}{ccc} m_{1} & m_{2} & 0\\ J_{x1} & J_{x2} & -m_{b}g\\ (l_{2}-L_{2})L_{1}/l_{1}l_{2} & L_{2}/l_{2} & 0 \end{array}\right]\left[\begin{array}{c} {\ddot{x}}_{1}(t)\\ {\ddot{x}}_{2}(t)\\ x_{b}(t) \end{array}\right]=\left[\begin{array}{c} -M{\ddot{X}}_{s}(t)-R_{x}(t)\\ -M(L_{f}+l_{b}){\ddot{X}}_{s}(t)-N_{y}(t)\\ {\ddot{X}}_{hd}(t)-{\ddot{X}}_{s}(t) \end{array}\right].$$

Then, the COM acceleration of the body is given by
(15)$${\ddot{x}}_{b}(t)=\frac{m_{1}}{m_{b}}{\ddot{x}}_{1}(t)+\frac{m_{2}}{m_{b}}{\ddot{x}}_{2}(t).$$

In the frontal plane, the COM displacement of the body is estimated by solving the following equations derived from (6), (7), and (13) for ${\ddot{y}}_{l}$, ${\ddot{y}}_{u}$, and *y_b_*:(16)$$\left[\begin{array}{ccc} 2m_{l}+m_{p}L_{l}/l_{l} & m_{u} & 0\\ J_{y1} & J_{y2} & -m_{b}g\\ (l_{u}-L_{u})L_{l}/l_{l}l_{u} & L_{u}/l_{u} & 0 \end{array}\right]\left[\begin{array}{c} {\ddot{y}}_{l}(t)\\ {\ddot{y}}_{u}(t)\\ y_{b}(t) \end{array}\right]=\left[\begin{array}{c} -M{\ddot{Y}}_{s}(t)-R_{y}(t)\\ -M(L_{f}+l_{b}){\ddot{Y}}_{s}(t)+N_{x}(t)\\ {\ddot{Y}}_{h}(t)-{\ddot{Y}}_{s}(t) \end{array}\right],$$

The COM acceleration of the body is given by
(17)$${\ddot{y}}_{b}(t)=\frac{2m_{l}l_{l}+m_{p}L_{l}}{m_{b}l_{l}}{\ddot{y}}_{l}(t)+\frac{m_{u}}{m_{b}}{\ddot{y}}_{u}(t).$$

These methods can be used to estimate the COM displacement and acceleration of the body in the sagittal and frontal planes. From these estimates, the COM velocity of the body can be estimated using the Kalman filter. Because the same technique was used for both the sagittal and frontal planes, the following explanation focuses only on the sagittal plane. The state and observation equations for applying the Kalman filter are defined as follows:(18)x(k)=Ax(k−1)+bu(k)+w(k),
(19)y(k)=cx(k)+v(k)
where *k* is the data number of the time series data, and
(20)$$x=\left[\begin{array}{c} x_{b}\\ {\dot{x}}_{b} \end{array}\right],\;\;y=x_{b},\;\;u={\ddot{x}}_{b},\;\;A=\left[\begin{array}{cc} 1 & \Delta t\\ 0 & 1 \end{array}\right],\;\;b=\left[\begin{array}{c} \Delta t^{2}/2\\ \Delta t \end{array}\right],\;\;c=\left[\begin{array}{cc} 1 & 0 \end{array}\right].$$

In (18)–(20), *Dt* is the sampling time, ***w*** is the process noise, and *v* is the observation noise. When ***Q_w_*** = ***ww****^T^* and *Q_v_* = *v*^2^, the COM position and velocity were estimated using the Kalman filter algorithm as follows:(21)$${\tilde{x}}^{-}(k)=Ax(k-1)+bu(k),$$
(22)$$P^{-}(k)=AP(k-1)A^{T}+Q_{w},$$
(23)$$G(k)=\frac{P^{-1}(k)c^{T}}{cP^{-}(k)c^{T}+Q_{v}},$$
(24)$$\tilde{x}(k)={\tilde{x}}^{-}\left(k\right)+G(k)(y(k)-c{\tilde{x}}^{-}(k)),$$
(25)$$P(k)=(I-G(k)c)P^{-}(k),$$
where $\tilde{x}$ is the estimated state vector, ***G*** is the Kalman gain, and ***P*** is the covariance matrix. The subscript ‘−’ means the prior prediction values.

The above estimation method requires body parameters such as mass, length, and moment of inertia. In this study, these were obtained from the subjects’ height and weight using specific formulae. The formulae were obtained from the literature [27] for the position of the center of gravity and moment of inertia of the body by segment and from the literature [28] for the length of the body. Table 1 shows the formulae for the body parameters used in this study.

## 3. Verification Methods

### 3.1. Experimental Protocol

To verify the accuracy of the anteroposterior (AP) and mediolateral (ML) COM estimation of the present methods, verification tests were implemented on 12 healthy male subjects (174.8 ± 4.7 cm, 61.6 ± 10.2 kg, 23.2 ± 3.8 years). The subjects performed six types of experiments as shown in Figure 2: (A) quiet standing, (B) ankle motion (AP): AP voluntary motion with the ankle joint strategy at 0.25 Hz, (C) ankle motion (ML): ML voluntary motion with the ankle joint strategy at 0.25 Hz, (D) hip motion (AP): AP voluntary motion with the hip joint strategy at 1 Hz, (E) support surface sway (AP): support surface sway for AP direction, (F) support surface sway (ML): support surface sway for ML direction. The experiments were conducted in order from (A) to (F). The frequencies of the movements in (B), (C), and (D) were set to ones that were easy to perform each movement. The subjects moved to the sound of the metronome.

The subjects stood barefoot with their feet positioned 15 cm between their heels and 16 cm between their thumbs. Their arms were down naturally, and their faces were facing forward. The measurement time for each experiment was 40 s, and six different motions were performed three times each.

### 3.2. Experimental Equipment

A force platform (TF-3040, Tec Gihan, Kyoto, Japan) and two IMUs (IMS-WD, Tec Gihan, Japan) were used in this experiment. The IMUs were attached to the back of the subject’s head and the support surface. The support surface was oscillated horizontally using a BASYS (Tec Gihan, Japan). The oscillation waveform was generated by the superposition of 20 harmonic waves at 0.05–1.0 Hz in increments of 0.05 Hz (Figure 3). An optical motion capture system (MAC3D System, Motion Analysis, Rohnert Park, CA, USA) was used to verify the accuracy of the proposed method. Twenty-nine reflective markers were attached to each subject according to the Helen–Hayes marker set, and four markers were placed on the force platform to derive the relative COM displacement of the body, lower body, and upper body from the support surface. The sampling frequency of all the instruments was set to 100 Hz.

### 3.3. Post-Processing

We applied a zero-phase high-pass filter (8th-order Butterworth filter) with a pass-frequency of 0.1 Hz to the horizontal forces *R_x_* and *R_y_* from the force platforms because the horizontal forces in quiet standing are much smaller than the vertical forces, and drift occurred in some data. The attitude angles of the IMU were derived from the three-axis acceleration and angular velocity data using an extended Kalman filter, and the horizontal head acceleration *Ẍ_h_* and *Ÿ_h_* were obtained by coordinate transformation. These values were used to estimate the displacement and velocity of the body COM and the acceleration of the lower and upper body COM.

For post-processing of the optical motion capture data, we applied a zero-phase low-pass filter (8th-order Butterworth filter) with a pass frequency of 3.0 Hz to the COM displacement of the body, lower body, and upper body and derived these velocities and accelerations by numerical differentiation. This study regarded the estimates from the optical motion capture system as true values.

We compared the following four COM estimation methods based on force platform measurements: (I) the estimation method based on a single inverted pendulum model using only a force platform, (II) the estimation method based on a double inverted pendulum model using a force platform and head IMU, (III) the COP filtering method, and (IV) the integration method for COM acceleration. Methods (III) and (IV) have been demonstrated in previous studies, and their details are provided in Appendix A. To eliminate high-frequency noise from the force platform and IMU measurements, the Kalman filter in (18)–(25) was applied to methods (I) and (II). The design parameters for the Kalman filter were set as ***Q_w_*** = diag(0.0025, 0.04), *Q_v_* = 1. Because only method (II) can estimate the COM accelerations of the lower and upper bodies, we evaluated the estimation accuracy of method (II).

The estimation accuracy of each method was evaluated using the root mean square error (RMSE) and Pearson’s correlation coefficient (C_C_) for the true values obtained from the optical motion capture system. The evaluation period was 30 s, ranging from 5 s to 35 s. The RMSE and C_C_ were calculated by
(26)$$\mathrm{RMSE}=\sqrt{\frac{1}{3000}\sum_{k=501}^{3500}{\left(q_{f}(k)-q_{m}(k)\right)}^{2}},$$
(27)$$C_{C}=\frac{\sum_{k=501}^{3500}\left(q_{f}(k)-{\bar{q}}_{f}\right)\left(q_{m}(k)-{\bar{q}}_{m}\right)}{\sqrt{\sum_{k=501}^{3500}{\left(q_{f}(k)-{\bar{q}}_{f}\right)}^{2}}\sqrt{\sum_{k=501}^{3500}{\left(q_{m}(k)-{\bar{q}}_{m}\right)}^{2}}},$$
where *q_f_* is the estimated value obtained from the force platform measurements, *q_m_* is the true value obtained from the optical motion capture measurements, ${\bar{q}}_{f}$ is the mean value of *q_f_*, and ${\bar{q}}_{m}$ is the mean value of *q_m_*.

## 4. Results

We compared the COM displacement and velocity of the body estimated using methods (I)–(IV) and the COM accelerations of the lower and upper bodies estimated using method (II) for motions (A)–(F) with the true values derived from optical motion capture measurements. The mean and standard deviation of the RMSE are listed in Table 2 (AP) and Table 3 (ML), and those of the C_C_ are listed in Table 4 (AP) and Table 5 (ML). The COM displacement and velocity of the body and COM accelerations of the lower and upper bodies obtained from motion capture (black line) and method (II) (blue line) for one subject (180 cm, 83 kg) for six motions, once each for (A) to (F), are plotted in Figure 4. The RMSE and C_C_ values for the same results are shown in the figures.

From the results in Table 2, Table 3, Table 4 and Table 5, the COP filtering method (III) had the highest estimation accuracy, except for the hip motion (AP), whereas method (II) had a higher estimation accuracy for the motion. Method (IV) was less accurate than method (II) for all the motions. Comparing methods (I) and (II), method (II) was more accurate for all motions, whereas the difference between the two methods was small, except for the hip motion. For hip motion, the estimation accuracy of method (I) was lower than that of method (II).

## 5. Discussion

For most motions, the COM estimation accuracies of methods (I)–(III) were relatively high. Method (III) was the most accurate except for the hip motion. However, few previous studies have applied method (III) to evaluate balance during quiet standing for two reasons. First, the method is empirical and not theoretical. As this method cuts off the higher frequencies of the COP, it is not possible to estimate the perturbation of the COM in the higher frequency range. Second, the COM obtained using method (III) depends on the COP via a low-pass filter. Therefore, the estimated COM is not suitable for balance evaluation compared with the COP.

Based on the above discussion, methods (I), (II), and (IV) should be adopted for balance evaluation. Among the three methods, method (II) had the highest COM estimation accuracy for all motions. However, except for hip motion, the difference in the estimation accuracy between methods (I) and (II) was minor. Although method (I) allows balance evaluation using only a force plate, it has a large estimation error for a large hip joint strategy motion. To distinguish the occurrence of estimation errors, we proposed a judgment technique based on the correlation coefficient *C_p_* between the COP and estimated COM in method (I). Comparing the *C_p_* between (A) and (D) in the sagittal plane, (A) was 0.952 ± 0.032, and (D) was 0.129 ± 0.399. The correlation coefficient was definitively lower in (D), indicating a lower estimation accuracy in method (I). Figure 5 shows the COM estimation results in the sagittal plane for the worst-case correlation coefficient (*C_P_* = 0.800) during quiet standing. This result indicates that method (I) is applicable for *C_p_* > 0.8.

Another advantage of method (II) is that it enables the estimation of COM accelerations of the lower and upper bodies. These accelerations should contribute to the discussion of ankle and hip joint strategies [29]. Focusing on the estimation accuracy of the COM acceleration for the upper and lower body, we noticed that the C_C_ of the upper body was low during hip motion in the sagittal plane, as shown in Table 4. To understand the reason for the time series waveform of the hip motion shown in Figure 4, we found that the acceleration of the upper body was considerably smaller than that of the lower body. Because the scale was smaller than the COM acceleration of the lower body, the C_C_ of the upper body was considered to decrease. Therefore, there is no problem with the estimation accuracy of the COM acceleration of the lower and upper bodies when using method (II).

The influence of the instruments on the accuracy of COM estimation should be considered in terms of force measurement precision and synchronization between devices. For the force measurement precision, there is a large difference between the large vertical and small horizontal forces during standing. This requires large amplification of the horizontal force and tends to cause drifts in the measurement signal at low frequencies. To avoid the influence of the drift, we applied a high-pass filter with a pass frequency of 0.1 Hz to the horizontal force signal. For synchronization between devices, method (II) requires precise synchronization because a synchronization error of approximately 20 ms should cause a large estimation error.

Methods (I) and (II), which allow online estimation of COM movement, can be easily applied to tests and rehabilitation. By applying the Kalman filter, the COM displacement and velocity can be estimated online with almost no phase delay after the noise removal. This feature should be useful in real-time display systems for the COM of balance tests [30,31,32] and real-time feedback systems for balance movements during interventions [33,34]. Combining the COM displacement and velocity estimated in Methods (I) and (II) allows flexible feedback intervention.

The limitations of this study were the lack of validation of differences in body shape, age, and sex and the consideration of vertical motion. The most significant problem with COM estimation methods based on equations of motion is body parameter errors [35]. As our validation test was limited to young men, the accuracy of the estimation methods for children, the elderly, and women was not sufficiently verified. Furthermore, the effect of vertical motion caused by hip and knee flexion on the estimation accuracy has not been verified.

## 6. Conclusions

In this study, we proposed COM estimation methods using a force platform and IMUs during standing and evaluated the estimation accuracy by comparing it with the optical motion capture system in various standing motions. Method (II), based on a double inverted pendulum model using a force platform and head IMU, had good estimation accuracy for all motions. Furthermore, this method can estimate the COM accelerations of the lower and upper bodies, which can be used to evaluate the ankle and hip joint strategies. Method (I), which uses only a force platform based on a single inverted pendulum model, also demonstrated good estimation accuracy, except for the hip joint strategy-dominant motion. The estimation error due to the hip strategy is acceptable if the correlation coefficient between the estimated COM displacement and the COP is greater than 0.8. Because the proposed method can estimate the displacement, velocity, and acceleration of the COM online, it can be used for the real-time display of the COM position and feedback intervention.

## 7. Patents

Kochi University of Technology, where the authors are affiliated, has applied for a patent related to this study in collaboration with Tec Gihan Co. Ltd. (Patent Application 2018-204854, Japan).

## Figures and Tables

**Figure 1 sensors-23-04933-f001:**
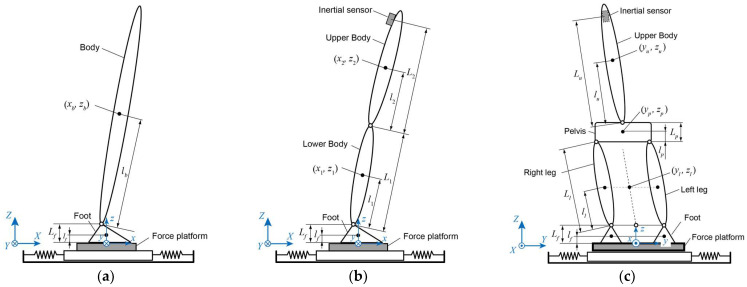
Rigid body link models used in this study: (**a**) single inverted pendulum model for the ankle joint strategy in the sagittal plane, (**b**) double-inverted pendulum model for the ankle and hip joint strategy in the sagittal plane, and (**c**) rigid link model for the balance motion in the frontal plane. All models allowed horizontal movement of the support surface.

**Figure 2 sensors-23-04933-f002:**
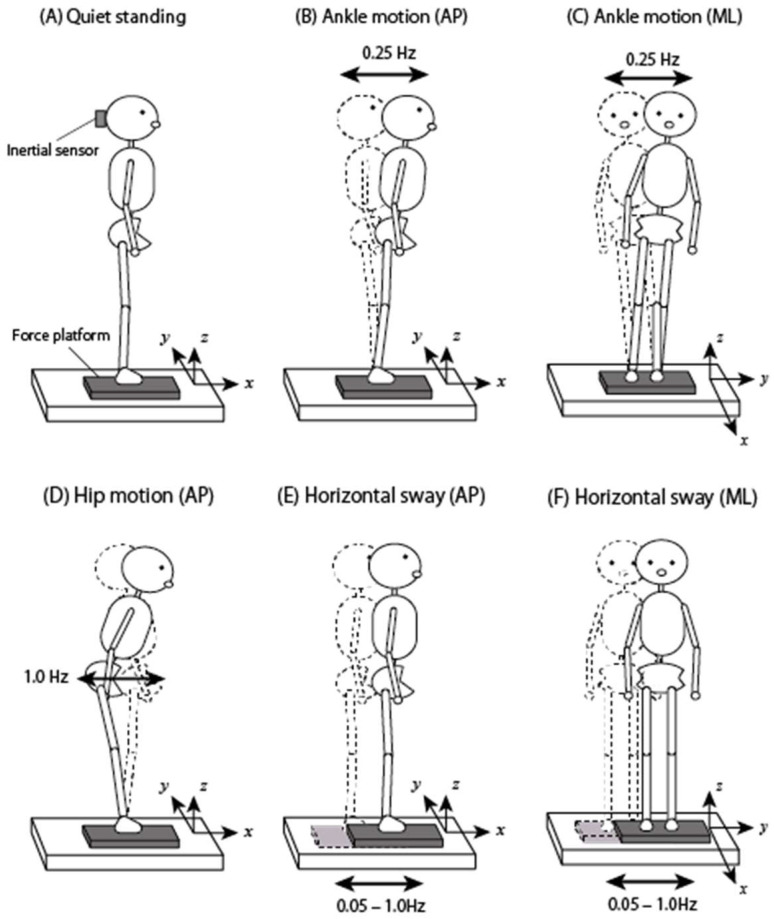
Six motions in an experiment to validate the accuracy of center of mass (COM) estimation: (**A**) quiet standing, (**B**) ankle joint strategy motion in the AP direction at 0.25 Hz, (**C**) ankle joint strategy motion in the ML direction at 0.25 Hz, (**D**) hip joint strategy motion in the AP direction at 1 Hz, (**E**) horizontal sway of the support surface in AP direction, and (**F**) horizontal sway of the support surface in the ML direction.

**Figure 3 sensors-23-04933-f003:**
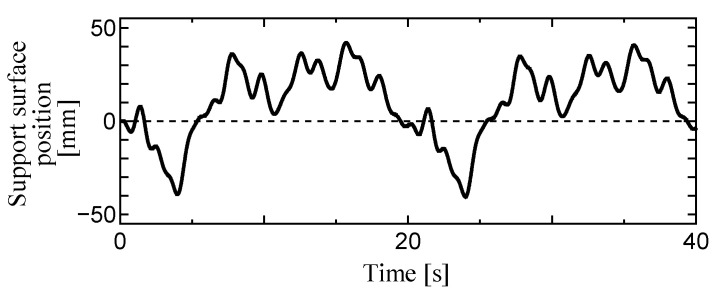
Time series waveforms of displacement of the support surface in tests (E) and (F).

**Figure 4 sensors-23-04933-f004:**
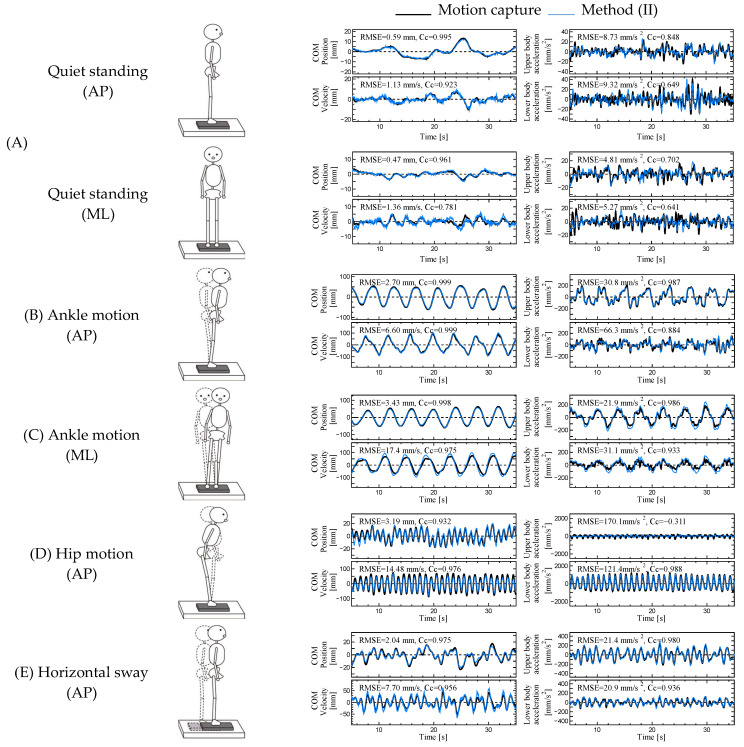
Time series data of COM displacements and velocities of the body and COM accelerations of the lower and upper bodies for six motions (**A**–**F**) for a subject (180 cm, 83 kg). The black and blue lines show the estimated values from the motion capture measurement and method (II), respectively.

**Figure 5 sensors-23-04933-f005:**
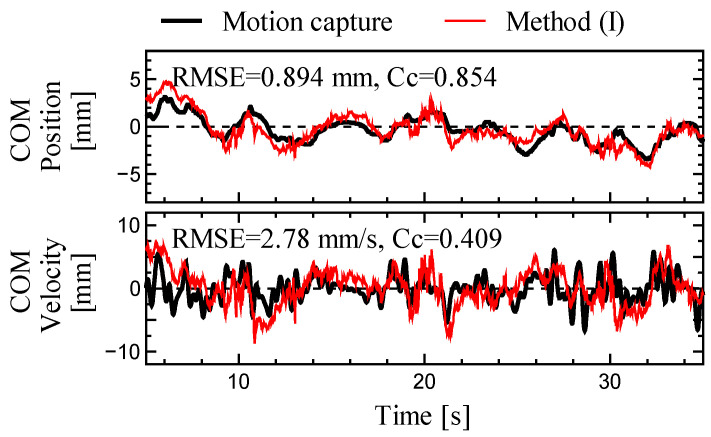
Time series data for the worst correlation coefficient *C_p_* in the sagittal plane (*C_p_* = 0.800). The black and red lines represent the estimates obtained from the motion capture system and method (I), respectively.

**Table 1 sensors-23-04933-t001:** Physical parameters of inverted pendulum models in the sagittal and frontal planes.

Sagittal Plane			Frontal Plane		
Segment	Symbol	Value	Segment	Symbol	Value
Body	*m_b_*	0.978 *M*	Legs	*m_l_*	0.161 *M*
	*J_b_*	0.0425 *MH*^2^		*J_l_*	0.00524 *MH*^2^
	*l_b_*	0.531 *H*		*l_l_*	0.285 *H*
Lower body	*m* _1_	0.322 *M*		*L_l_*	0.460 *H*
	*J* _1_	0.00223 *MH*^2^	Pelvis	*m_p_*	0.187 *M*
	*l* _1_	0.285 *H*		*l_p_*	0.056 *H*
	*L* _1_	0.460 *H*		*L_p_*	0.144 *H*
Upper body	*m* _2_	0.656 *M*	Upper body	*m_u_*	0.469 *M*
	*J* _2_	0.0114 *MH*^2^		*J_u_*	0.00714 *MH*^2^
	*l* _2_	0.191 *H*		*l_u_*	0.109 *H*
	*L* _2_	0.434 *H*		*L_u_*	0.290 *H*
Foot	*L_f_*	0.038 *H*	Foot	*L_f_*	0.038 *H*

*M*: Body weight, *H*: Height.

**Table 2 sensors-23-04933-t002:** Root mean square error (RMSE) between the estimates of the four methods (I)–(IV) and the true values obtained from the optical motion capture system for the COM displacements and velocities of the body and COM accelerations of the lower and upper bodies in the sagittal plane.

Motion	Method	COM Position	COM Velocity	Lower Body Acceleration	Upper Body Acceleration
		mm	mm/s	mm/s^2^	mm/s^2^
(A) Quiet Standing	RMS	3.25 ± 1.09	2.36 ± 0.59	10.2 ± 4.2	11.0 ± 4.1
	(I)	0.65 ± 0.21	1.83 ± 0.56	-	-
	(II)	0.59 ± 0.18	1.75 ± 0.55	9.6 ± 4.0	8.3 ± 4.1
	(III)	0.31 ± 0.12	0.72 ± 0.44	-	-
	(IV)	1.03 ± 0.67	3.71 ± 1.18	-	-
(B) Ankle Motion (AP)	RMS	26.89 ± 7.70	33.70 ± 9.81	48.2 ± 13.0	85.6 ± 24.2
	(I)	2.89 ± 1.45	9.73 ± 3.86	-	-
	(II)	2.62 ± 1.17	8.86 ± 3.77	33.7 ± 8.3	27.9 ± 8.9
	(III)	1.50 ± 0.43	2.29 ± 0.52	-	-
	(IV)	9.32 ± 6.96	10.81 ± 3.04	-	-
(D) Hip Motion (AP)	RMS	9.03 ± 2.64	35.65 ± 14.50	421.5 ± 202.9	119.3 ± 66.5
	(I)	15.10 ± 6.86	32.59 ± 15.28	-	-
	(II)	3.47 ± 1.67	12.16 ±5.36	100.0 ± 37.4	103.9 ± 57.7
	(III)	4.44 ± 2.06	24.01 ± 11.28	-	-
	(IV)	5.17 ± 3.34	10.22 ± 6.27	-	-
(E) Horizontal Sway (AP)	RMS	10.89 ± 2.44	16.36 ± 1.76	50.3 ± 11.6	99.2 ± 11.3
	(I)	1.99 ± 0.44	7.44 ± 0.81	-	-
	(II)	1.60 ± 0.34	6.90 ± 0.79	30.6 ± 8.4	32.2 ± 8.8
	(III)	1.21 ± 0.24	2.33 ± 0.50	-	-
	(IV)	12.15 ± 7.24	15.09 ± 3.92	-	-

**Table 3 sensors-23-04933-t003:** RMSE between the estimates of the four methods (I)–(IV) and the true values obtained from the optical motion capture system for the COM displacements and velocities of the body and COM accelerations of the lower and upper bodies in the frontal plane.

Motion	Method	COM Position	COM Velocity	Lower Body Acceleration	Upper Body Acceleration
		mm	mm/s	mm/s^2^	mm/s^2^
(A) Quiet Standing	RMS	1.51 ± 0.62	1.55 ± 0.53	9.4 ± 4.9	10.4 ± 6.1
	(I)	0.56 ± 0.15	1.76 ± 0.46	-	-
	(II)	0.56 ± 0.15	1.75 ± 0.45	8.7 ±4.9	9.3 ± 6.3
	(III)	0.17 ± 0.07	0.66 ± 0.44	-	-
	(IV)	2.48 ± 2.78	5.23 ± 2.20	-	-
(C) Ankle Motion (ML)	RMS	34.47 ± 8.77	42.86 ± 10.47	53.1 ±11.9	93.7 ± 21.4
	(I)	3.60 ± 2.50	20.93 ± 5.38	-	-
	(II)	3.33 ± 1.96	19.67 ± 5.11	34.6 ±8.1	32.2 ± 8.4
	(III)	1.41 ± 0.80	2.16 ± 1.05	-	-
	(IV)	25.95 ± 22.62	33.36 ± 9.14	-	-
(F) Horizontal Sway (ML)	RMS	10.08 ± 1.98	19.10 ± 1.95	61.0 ±8.0	100.0 ± 11.9
	(I)	2.32 ± 0.41	7.95 ± 1.16	-	-
	(II)	1.79 ± 0.36	7.38 ± 1.28	23.8 ±3.4	29.9 ± 5.8
	(III)	1.10 ± 0.24	1.91 ± 0.28	-	-
	(IV)	12.36 ± 7.71	16.39 ± 3.78	-	-

**Table 4 sensors-23-04933-t004:** Correlation coefficients (Cc) between the estimated values from methods (I)–(IV) and the true values obtained from the optical motion capture system for COM displacement and velocity of the body and COM acceleration of the lower and upper bodies in the sagittal plane.

Motion	Method	COM Position	COM Velocity	Lower Body Acceleration	Upper Body Acceleration
(A) Quiet Standing	(I)	0.978 ± 0.023	0.778 ± 0.122	-	-
	(II)	0.981 ± 0.020	0.781 ± 0.126	0.377 ± 0.136	0.446 ± 0.189
	(III)	0.997 ± 0.006	0.951 ± 0.073	-	-
	(IV)	0.954 ± 0.031	0.655 ± 0.152	-	-
(B) Ankle Motion (AP)	(I)	0.998 ± 0.002	0.980 ± 0.024	-	-
	(II)	0.998 ± 0.001	0.980 ± 0.025	0.689 ± 0.126	0.953 ± 0.031
	(III)	0.999 ± 0.001	0.999 ± 0.001	-	-
	(IV)	0.943 ± 0.060	0.970 ± 0.026	-	-
(D) Hip Motion (AP)	(I)	0.102 ± 0.363	0.767 ± 0.071	-	-
	(II)	0.938 ± 0.046	0.949 ± 0.043	0.958 ± 0.074	0.377 ± 0.429
	(III)	0.894 ± 0.063	0.893 ± 0.129	-	-
	(IV)	0.844 ± 0.183	0.942 ± 0.082	-	-
(E) Horizontal Sway (AP)	(I)	0.984 ± 0.008	0.934 ± 0.013	-	-
	(II)	0.989 ± 0.006	0.945 ± 0.011	0.813 ± 0.056	0.935 ± 0.028
	(III)	0.995 ± 0.002	0.992 ± 0.003	-	-
	(IV)	0.711 ± 0.196	0.840 ± 0.082	-	-

**Table 5 sensors-23-04933-t005:** Cc between the estimated values from methods (I)–(IV) and the true values obtained from the optical motion capture system for COM displacement and velocity of the body and COM acceleration of the lower and upper bodies in the frontal plane.

Motion	Method	COM Position	COM Velocity	Lower Body Acceleration	Upper Body Acceleration
(A) Quiet Standing	(I)	0.922 ± 0.062	0.636 ± 0.131	-	-
	(II)	0.920 ± 0.068	0.623 ± 0.135	0.252 ± 0.125	0.339 ± 0.151
	(III)	0.991 ± 0.010	0.889 ± 0.100	-	-
	(IV)	0.713 ± 0.267	0.452 ± 0.159	-	-
(C) Ankle Motion (ML)	(I)	0.998 ± 0.001	0.974 ± 0.017	-	-
	(II)	0.998 ± 0.001	0.971 ± 0.019	0.872 ± 0.075	0.962 ± 0.034
	(III)	1.000 ± 0.000	0.999 ± 0.001	-	-
	(IV)	0.919 ± 0.120	0.975 ± 0.022	-	-
(E) Horizontal Sway (ML)	(I)	0.974 ± 0.011	0.946 ± 0.019	-	-
	(II)	0.985 ± 0.006	0.951 ± 0.022	0.924 ± 0.023	0.957 ± 0.022
	(III)	0.995 ± 0.002	0.996 ± 0.002	-	-
	(IV)	0.710 ± 0.195	0.873 ± 0.073	-	-

## Data Availability

doi: https://doi.org/10.6084/m9.figshare.22584100.v2. Description: The displacement, velocity, and acceleration of the body COM and the COM of the lower and upper bodies estimated from a motion capture and a force platform were saved as CSV files. The correlation coefficients and root mean square errors between the motion capture and each method were also stored.

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
