# Peer review of "Center of Mass Estimation Using a Force Platform and Inertial Sensors for Balance Evaluation in Quiet Standing"

_sensors, 2023, doi:10.3390/s23104933_

Round 1

Reviewer 1 Report

General Comments

The aim of this study was to compare the COM estimation accuracy of the proposed method with those of other methods in previous studies, using motion capture system as the reference value. Accurate estimation of the center of mass (COM) is crucial for evaluating balance control. The paper is generally well written, but the following are specific comments that could further enhance the study methodology.

Specific Comments

Introduction

·       Previous COM estimation techniques were introduced (lines 46 – 53). However, the disadvantages of these techniques were not highlighted in the introduction, although it was briefly highlighted in the abstract. This could lead to a better understanding as to why a new alternative method is required and needed in the study.

Validation Methods

·       Is there a power analysis conducted for a sample size of 12?

·       What is the age of the participants?

·       How is the frequency of (B) and (C) (0.25 Hz) controlled or determined?

·       Were trial conditions randomized?

·       Were there practice trials before actual data recording?

·       Line 188 – What type of filter was used? Was it butterworth?

·       For analysis, root mean square error (RMSE) and Pearson’s correlation coefficient (CC) were used? However, no statistical analysis or p value was used. How could results be determined to be ‘significantly lower’ in results (line 222). With no statistical comparison, it is perhaps less convincing that one method works better over the other. External validity could also be questionable. Could ANOVA with repeated measures or p values of CC be provided? Or could Bland-Altman analysis be used?

Discussion and Conclusion

·       For better flow, limitations should be shifted to the end of discussion.

A conclusion should also be added. 

Author Response

Thank you for reviewing our manuscript. Please see the attachment.

Reviewer 2 Report

Peer Review Report

Ms. Ref. No.: sensors-2387115

Title: Center of mass estimation using a force platform and inertial sensors for balance evaluation in quiet standing

Authors: Motomichi Sonobe and Yoshio Inoue

The subject of the article is within scope of the journal. The subject presented in the manuscript is very interesting. However, the manuscript needs to be improved in my opinion. I recommend the paper for major revision. I believe that the authors will find below some suggestions, which will help them to improve their manuscript (I hope so):

Major comments:

1)      In the Introduction, the authors should take into account also that some authors propose also a mechanical model for prediction of the fall [DOI: 10.3390/app13085068]. Furthermore, the introduction of the manuscript should be significantly extended, in particular the authors should explain how the center of mass of the human body is estimated in the literature (not only in the case of balance study – in general, what are typically used models?). Description of the Kalman filter in the introduction and examples of its application in biomechanical studies should be also given in the Introduction.

2)      The Kalman filter was used only in the second method or in the both methods? If in both methods it should be described in a separate subsection. Furthermore, the description of Kalman filter should be extended and the quantities in Eq. (12) should be explained in more details?

3)      What is the motivation for using the Kalman filter? What are alternatives for that?

4)      The authors can compare the results with and without using Kalman filter.

5)      Line 183, how the authors choose the values of Qw and Qv? It should be explained. What would be obtained if the authors choose another values?

6)      In Eqs. (1)-(10) maybe it would be good to add time in order to highlight that it is continuous and depends on time (like in Appendix). What is k in Eqs. (11)-(12)?

7)      Lines 146-148. The authors should give more details how exactly they obtained the values in the Table.

8)      References are not numbers, but itemized.

Minor comments:

9)      Do not introduce abbreviation in the abstract of the manuscript. Simply use the whole terms.

10)  Improve and unify a punctuations in equations. For instance, it is better to use commas separately after each equation than after two equations together (see for examples Eq. (1) and others). In many equations commas should be added.

11)  Number each equation in the manuscript.

12)  Lack of index 2 in Eq. (8).

13)  Is it needed to write “the body COM” instead of simply “the COM”?

Conclusion:

The subject of the paper and the manuscript are very interesting. I recommend the manuscript for major revision.

Author Response

(The authors gave the same response as above.)

Round 2

Reviewer 2 Report

The authors answered all my comments. Now, I can recommend the manuscript for publication in Sensors.